# De novo Drug Design using Reinforcement Learning with Dynamic Vocabulary

## Abstract

*De novo* drug design constitutes a fundamental challenge within the domain of computer-aided drug discovery (CADD). Generative models relying on SMILES molecular strings have emerged as promising tools for this purpose. However, extant SMILES-based generative models all adopt a fixed vocabulary, leading to deficiencies in both sampling efficiency and interpretability. In this paper, we propose RLDV, a reinforcement learning (RL) algorithm based on a GPT agent, which uses a dynamic chemical vocabulary (DV) during RL iterations. Specifically, we utilize SMILES pair encoding to analyze high-scoring molecular SMILES strings generated during the RL process, and extract their high-frequency common substrings, which are then added as new tokens to the agent's vocabulary. These additions aid in the generation of molecules during subsequent RL steps. Experimental results on the GuacaMol benchmark demonstrate that our algorithm outperforms existing models across multiple tasks, highlighting the practical significance of the dynamic vocabulary in drug design. Furthermore, the application of our algorithm in the design of protein-targeting drugs for SARS-CoV-2 underscores its substantial practical relevance.

## 1 Introduction

Over the past few years, the application of artificial intelligence (AI) in computer-aided drug discovery (CADD) has witnessed remarkable progress (Sabe et al., 2021; Zhang et al., 2023). This advancement has been driven by the continual accumulation of data from diverse scientific disciplines such as biology, chemistry, and pharmacy, coupled with ongoing enhancements in algorithmic capabilities. Consequently, there has been a revolutionary improvement in the efficiency of drug discovery processes, notably achieving performance levels that satisfy pharmaceutical researchers in tasks such as virtual screening (Maia et al., 2020) and protein structure prediction (Jumper et al., 2021). Nevertheless, as of the present, *de novo* drug molecular design remains a paramount challenge within the domain of CADD. Despite the introduction of numerous methodologies including deep learning aimed at this challenge, their practical efficacy in real-world drug development endeavors remains limited (Dara et al., 2022).

Due to the success of generative models in the field of natural language processing, SMILES-based algorithms for *de novo* drug design have been regarded as promising, particularly in light of the potential demonstrated by generative pre-training and RL-based fine-tuning in the chemical domain. However, current SMILES-based generative models all employ fixed vocabularies, with the majority consisting solely of atomic-level tokens, encompassing atoms, ions, functional groups, numbers, and symbols. This vocabulary is concise, requiring fewer than 200 tokens to generate SMILES strings for nearly all small molecules, but it inherently lacks any chemical structural prior knowledge, and maximizes the number of tokens required to make up a SMILES string. On the other hand, some recent approaches employ larger fixed vocabularies that include pre-extracted SMILES substrings from datasets (Li et al., 2023). However, such vocabularies often encompass thousands or more tokens, containing a plethora of information that is unhelpful for the particular molecular generation tasks at hand.

In response to the aforementioned challenges, drawing inspiration from the prevalent fragment-based approaches in molecular graph generation, we introduce RLDV (Reinforcement Learning with Dynamic Vocabulary), which innovatively integrates a dynamic vocabulary module into the

reinforcement learning framework based on SMILES representations. In particular, the vocabulary of the agent in RLDV comprises two components: a fixed set of atomic-level tokens and a dynamically updated set of higher-level tokens. Throughout the RL process, we employ the SMILES pair encoding algorithm to extract high-frequency common sub-strings from the generated high-scoring molecules. These sub-strings are then periodically integrated into the dynamic part of the agent's vocabulary as tokens. The dynamic vocabulary explicitly preserves task-specific structural information at the SMILES level. This not only enhances the efficiency of subsequent RL steps but also enhances the model's interpretability.

Experimental results on the GuacaMol benchmark demonstrate that RLDV outperforms existing baselines, and visual analysis of the tokens in the dynamic vocabulary reveals that the extracted substrings correspond to substructures within the target molecules, effectively boosting the model's generation efficiency. Furthermore, in experiments on the design of inhibitors against SARS-CoV-2 protein targets, RLDV exhibits potential in real-world drug discovery. The appearance of crucial substructures associated with specific protein binding within the dynamic vocabulary reflects the task-specific interpretability of our algorithm. Our code will be publicly available following the publication of this paper.

## 2 RELATED WORKS

### 2.1 DE NOVO DRUG DESIGN

In some tasks within the field of CADD, machine learning methods have demonstrated practical success, such as in molecular docking simulation (Trott & Olson, 2010), molecular property prediction (Wieder et al., 2020), protein structure prediction (Jumper et al., 2021), and retro-synthesis (Segler et al., 2018b; Liu et al., 2023). However, the ultimate goal of drug discovery is the production of real-world drugs, with the most critical step being the design of candidate drug compounds that meet specified criteria. Presently, existing approaches cannot offer satisfactory solutions to this challenge, which arises from the vast and unstructured chemical space (Polishchuk et al., 2013), as well as the complexity of the relationship between molecular biochemical properties and structures.

As a subset of molecular generation, *de novo* drug design algorithms fundamentally rely on molecular representations. From this perspective, these algorithms can be primarily categorized into three classes: those based on 1D molecular strings, those based on 2D molecular graphs, and those based on 3D geometric structures (Du et al., 2022). In 2D and 3D algorithms, some cutting-edge machine learning techniques, such as diffusions (Xu et al., 2022; Vignac et al., 2023), flows (Bengio et al., 2021), and equivariant networks (Gebauer et al., 2019; Adams & Coley, 2022), have found widespread applications. However, up to the present, for *de novo* drug design, reinforcement learning methods based on 1D strings (Olivecrona et al., 2017) and genetic algorithms based on 2D graphs (Jensen, 2019) remain the most competitive choices (Gao et al., 2022). With the explosive developments of generative AI in the field of natural language processing (NLP) in recent years (Vaswani et al., 2017; Devlin et al., 2019; Brown et al., 2020), we believe that methods based on 1D molecular strings hold significant potential in the context of *de novo* drug design.

**SMILES-based molecular generation**  SMILES (Simplified Molecular Input Line Entry System) (Weininger, 1988) is the most widely utilized molecular string representation. It encodes 2D molecular graphs into concise and readable character sequences, substantially aiding the field of chemoinformatics in the processing of molecular data. Currently, numerous machine learning techniques have been employed for the SMILES-based generation of molecules, including recurrent neural networks (RNNs) Segler et al. (2018a), variational autoencoders (VAEs) (Gómez-Bombarelli et al., 2018; Eckmann et al., 2022), generative adversarial networks (GANs) (Guimaraes et al., 2017), genetic algorithm (GA) (Yoshikawa et al., 2018), and Bayesian optimization (BO) (Moss et al., 2020). Notably, Transformer models have demonstrated their capabilities in processing and generating SMILES strings (Bagal et al., 2022; Irwin et al., 2022; He et al., 2022).

**RL-based *de novo* drug design**  Reinforcement Learning (RL) is a machine learning paradigm that enables intelligent agents to maximize cumulative rewards through interactions with an environment. Currently, it has gained significant popularity in the field of *de novo* drug molecular design. The Reinvent framework (Olivecrona et al., 2017; Blaschke et al., 2020a) pioneers the use

of a SMILES-based deep reinforcement learning algorithm to train an RNN model for generating SMILES strings, and now it continues to lead the way in *de novo* drug design (Gao et al., 2022). Building upon Reinvent, several RL-based methods have introduced techniques such as curriculum learning (Mokaya et al., 2023), knowledge distillation (Wang et al., 2021), alternating rewards (Goel et al., 2021), and RNN-based property predictor (Popova et al., 2018) for drug design. Furthermore, some graph-based approaches also employ RL techniques, typically treating the addition or removal of molecular components like atoms, bonds, and cycles as actions to intuitively design molecular graphs, as exemplified by methods like GCPN (You et al., 2018), MolDQN (Zhou et al., 2019), RationaleRL (Jin et al., 2020), GEGL (Ahn et al., 2020), and FREED (Yang et al., 2021).

**Fragment-based drug design** The utilization of molecular fragments, structural motifs, subgraphs, or substructures in drug design, as opposed to only using fundamental atomic-level components, as exemplified by methods such as RationaleRL (Jin et al., 2020), FREED (Yang et al., 2021), RS-VAE (Kong et al., 2022), and MiCaM (Geng et al., 2023), presents several advantages. Firstly, assembling the same molecule can be achieved through fewer steps, reducing the combinatorial complexity associated with molecular generation, thereby enhancing algorithm performance and efficiency. Secondly, fragments and motifs explicitly retain task-specific information, thereby improving the model's interpretability. Thirdly, we can inject prior knowledge of certain structures into the set of substructures, facilitating the utilization of expert experience. However, the application of molecular fragments is primarily confined to 2D graph-based *de novo* molecular design, while 1D string-based methods have not yet fully exploited this concept.

## 2.2 DYNAMIC VOCABULARY

In scenarios involving the application of language models where tokens not present in the pre-trained dataset are encountered, it becomes imperative to dynamically adjust or expand the model's vocabulary. This dynamic vocabulary serves to endow the model with heightened adaptability and accuracy, thereby mitigating the maintenance costs arising from sparse data while augmenting its capacity to provide personalized user support and multilingual support. Consequently, dynamic vocabularies have found extensive utilization in domain-specific applications with many custom terminologies, in the processing of user-generated content, and within multilingual systems Jean et al. (2015); Wu et al. (2018); Lakew et al. (2018); Amba Hombaiah et al. (2021). Notably, it is worth mentioning that Lan et al. (2023) introduces a method for generating new text by copying text segments from an existing text collection. This approach fundamentally relies on a dynamic vocabulary and demonstrates significant domain-specific adaptability.

Existing algorithms for *de novo* drug design all utilize fixed vocabularies, whereas we are the first to introduce a dynamic vocabulary for tokenizing SMILES strings, thereby significantly enhancing the model's sampling efficiency and task-specific interpretability.

## 3 METHODOLOGY

In this section, we will provide a detailed exposition of our *de novo* drug design algorithm, RLDV (Reinforcement Learning with Dynamic Vocabulary), including reinforcement learning (RL) based on a GPT agent, dynamic chemical vocabulary, and other technical details.

### 3.1 REINFORCEMENT LEARNING WITH A GPT AGENT

We construct a reinforcement learning framework for designing drug candidates with specific properties. As illustrated in Figure 1, for each goal-directed molecular generation task, the RL process of RLDV consists of iterative steps, wherein the parameters of a GPT agent with a dynamic vocabulary is updated using the REINFORCE algorithm (Williams, 1992). In specific terms, a GPT prior model pre-trained on a large chemical language dataset (see section 3.3) is utilized to initialize the agent. In each RL step, the GPT agent generates a set of SMILES strings, and the likelihood of these strings from the prior model as well as their scores predicted by the scoring function are used to compute the loss for updating the agent.

Furthermore, a memory is maintained to store the top-$k$ high-scoring molecules in their canonical SMILES representations. In each RL step, newly generated molecules are used to update this mem-

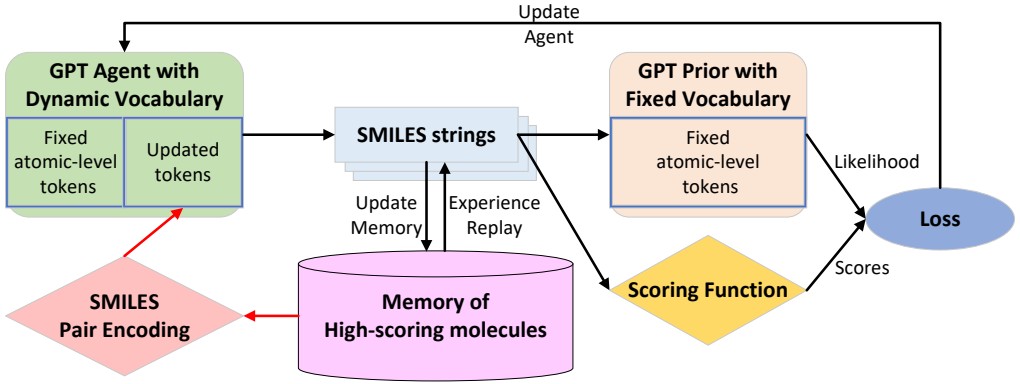

Figure 1: Overview of the RL framework of RLDV.

ory, and some high-scoring molecules are randomly sampled for experience replay. Additionally, at regular intervals of steps, new tokens are generated from these high-scoring molecules by SMILES pair encoding algorithm, which are used to update the dynamic vocabulary of the RL agent.

The GPT agent's vocabulary consists of two parts: one comprises fixed atomic-level tokens initialized from the prior's vocabulary, and the other comprises higher-level tokens that are dynamically updated during the RL process. In contrast, the vocabulary and parameters of the prior model remain unchanged during the RL process.

Typically, the number of tokens in the agent's vocabulary tends to increase during the RL process, indicating the gradual accumulation of task-specific knowledge within the vocabulary. Moreover, the average scores of the molecules generated by the GPT agent show an upward trend, suggesting that it progressively acquires a understanding of the structural patterns of desirable molecules in the chemical language.

**Loss function** In RLDV, we define the scores predicted by the task-specific scoring function as the RL reward, as our primary objective is to enhance the scores of molecules generated by the GPT agent. Additionally, inspired by Olivecrona et al. (2017), to prevent the agent from losing the fundamental grammar of chemical language learned during the pre-training phase (ensuring the validity of generated SMILES strings), we penalize the deviation between the policies of agent and prior within the loss function. It is worth noting that because the vocabulary of the agent undergoes dynamic changes during the RL process, the tokenization results for the same SMILES string may differ between the agent and the prior. The loss function used to update the agent is defined as follows:

$$L(x; \Theta) = \big[\log P_{\mathrm{Prior}}(T_{\mathrm{Prior}}(x)) - \log P_{\mathrm{Agent}}(x) + \sigma \cdot s(x)\big]^2 \qquad (1)$$

where $\Theta$ represents the parameters of the GPT agent, $x$ is a generated SMILES string, $\sigma$ is a coefficient for controlling the term of scores, $s(\cdot)$ is the task-specific scoring function, $T_{\mathrm{Prior}}(\cdot)$ refers to the atomic-level tokenizer of the prior model, and $P_{\mathrm{Prior}}(\cdot)$ and $P_{\mathrm{Agent}}(\cdot)$ respectively calculates the likelihood of generating a series of tokens from $\mathrm{Prior}$ and $\mathrm{Agent}$.

### 3.2 DYNAMIC CHEMICAL VOCABULARY

The existing SMILES-based molecular generation models utilize fixed vocabularies, thereby limiting the efficiency and interpretability of the models. In RLDV, we introduce a novel concept of dynamic vocabulary to the chemical language. The dynamic vocabulary is updated dynamically during the RL process, thereby explicitly preserving the task-specific knowledge acquired by the agent within the new tokens added to the vocabulary, which also contribute to subsequent RL steps. Regarding the vocabulary updating mechanism, we employ the SMILES pair encoding algorithm to extract frequent substrings from SMILES strings of high-scoring molecules. These substrings are merged from atomic-level tokens and are used for updating the vocabulary of the RL agent. Figure 2 provides an illustrative example of updating the dynamic vocabulary.

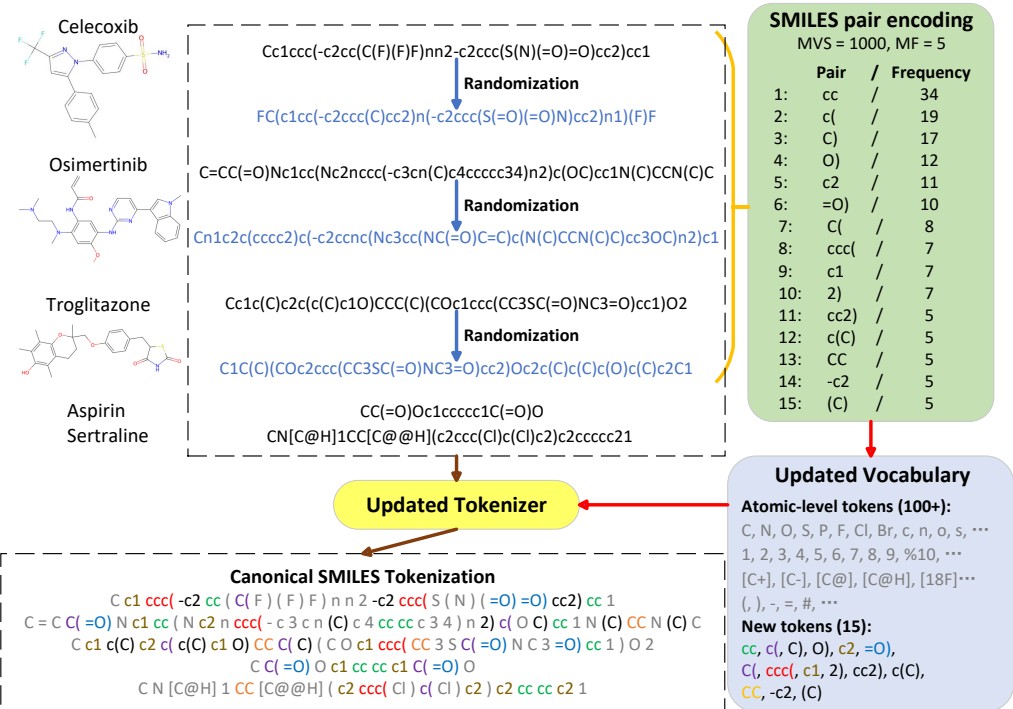

Figure 2: An illustrative example of updating the dynamic vocabulary. We use canonical SMILES strings of 3 drugs as the training set for SMILES pair encoding, incorporating one time of SMILES randomization. Employing a sufficiently large MVS of 1000 and a MF of 5, we extract 15 high-frequency substrings, which are used to update the atomic-level vocabulary. We demonstrate the tokenization results of applying the updated vocabulary to the training set and new SMILES strings. Atomic-level tokens are depicted in gray, tokens exclusive to the training set are shown in black, and tokens present in both the training set and new strings are represented in color. The results indicate that the new vocabulary effectively encodes the training set and exhibits representation ability for new SMILES strings to some degree.

### 3.2.1 SMILES PAIR ENCODING

In the pursuit of identifying the most frequent substrings in a set of SMILES strings, we introduce Byte Pair Encoding (BPE) (Gage, 1994) to the chemical language, which is a commonly used technique in NLP for data compression and text tokenization (Sennrich et al., 2016). The core idea behind BPE is to construct a smaller or more effective vocabulary by merging or encoding a frequently occurring pair of characters into a single token.

With SmilesPE (Li & Fourches, 2021) as a reference, our SMILES pair encoding algorithm takes a set of SMILES strings as input and aims to expand the chemical vocabulary with common SMILES substrings in the set. Specifically, we initialize from the atomic-level vocabulary, which is tokenized from ChEMBL (Mendez et al., 2019), a large dataset of drug molecules. Then for the given set of SMILES strings, through an iterative process, the occurrence of all token pairs in the tokenized set are counted. The token pair with the highest frequency of occurrence is merged as a new token, which is then incorporated into the evolving vocabulary. This iterative procedure terminates upon the satisfaction of either of two conditions: (1) the attainment of a pre-determined maximum vocabulary size (MVS) or (2) the absence of any token pair with a frequency exceeding a pre-defined minimum frequency (MF). The MVS and MF are two pivotal hyper-parameters for the SMILES pair encoding algorithm.

Moreover, for a given molecule, distinct atom orderings lead to various equivalent SMILES strings, each of which encompasses different substrings. Therefore, we employ SMILES randomization

(enumeration) (Bjerrum, 2017) as a data augmentation technique in SMILES pair encoding algorithm, which is a widely-adopted strategy in SMILES-based deep learning algorithms (Arús-Pous et al., 2019; Chen & Tseng, 2021).

### 3.2.2 VOCABULARY UPDATE

With the SMILES pair encoding algorithm in place, following several RL iterations, we are able to obtain high-frequency common substrings of high-scoring molecules at the SMILES level. These substrings are formed by concatenating two or more atomic-level tokens, preserving task-specific knowledge in the form of short sequences of tokens, which will aid in the subsequent RL steps.

Specifically, after each vocabulary update interval (VUI), we apply the SMILES pair encoding algorithm to the high-scoring molecules stored in the memory, with a sufficiently large maximum vocabulary size (MVS) and a minimum frequency (MF) not less than 100. Consequently, we typically obtain several hundred new higher-level tokens, which are used to replace the non-fixed part of the GPT agent's vocabulary. In other words, at each interval, the higher-level part of tokens in the agent's vocabulary are entirely refreshed, and vocabulary accumulated from previous steps may also be discarded in subsequent steps.

After each update, tokens corresponding to the same code in the GPT agent's vocabulary may change, thus the vocabulary update interval (VUI) should not be too short to ensure that agents adequately learn and utilize new tokens to generate new molecular strings. Additionally, it is worth noting that higher-level tokens are composed of atomic-level tokens, both of which exist in the agent's vocabulary, resulting in the possibility of multiple generated paths for the same SMILES string within the agent. We anticipate reinforcing newly discovered higher-level tokens in this regard, making experience replay (see section 3.3) indispensable.

### 3.3 OTHER TECHNICAL DETAILS

**Pre-training GPT for SMILES generation** Inspired by MolGPT (Bagal et al., 2022), we employ a tiny GPT-2 model (Radford et al., 2019) as a generator for SMILES strings. We construct a vocabulary comprising 108 atomic-level tokens, while setting the maximum vocabulary size (MVS) of the model to 1000 to accommodate the storage of higher-level tokens effectively. Pre-training is conducted using the ChEMBL dataset Mendez et al. (2019), wherein SMILES strings containing tokens outside the vocabulary and those exceeding a length of 100 tokens are removed. The pre-processed dataset encompasses approximately 1.8 million molecules. Through supervised training together with SMILES randomization, we obtain a GPT prior model capable of generating SMILES strings with a validity exceeding 98%.

**Memory & Experience replay** Inspired by Blaschke et al. (2020b), we utilize a memory that sorts and stores top-$k$ high-scoring molecules discovered during the RL process, with $k$ set to 1000. This approach not only facilitates vocabulary updates and the final output of drug design results, but also enables experience replay (Lin, 1992), a commonly used technique in the field of reinforcement learning, for high-scoring molecules. We define a hyper-parameter $t$, named ER, to control the number of molecules for experience replay. In each RL step, $t$ molecules are randomly sampled from the top-$5t$ high-scoring molecules, and they are then used to update the agent and reinforce its learning of higher-level tokens.

## 4 EXPERIMENTS

### 4.1 GUACAMOL BENCHMARK

### 4.1.1 EXPERIMENTAL SETUP

GuacaMol (Brown et al., 2019) is a widely recognized benchmark for *de novo* drug design, containing 20 meticulously designed goal-directed molecular generation tasks. These tasks cover a wide range of objectives, including designing similar molecules, rediscovering specific structures, enumerating isomers, designing median molecules, and some multi-property objectives. Therefore, GuacaMol can comprehensively evaluate the performance of models for *de novo* drug design. Each

task assess a model with a score ranging from 0 to 1, where a higher score indicates superior performance.

We evaluate our RLDV algorithm on all 20 goal-directed tasks in GuacaMol, with each task running for 5000 RL steps, which is chosen to ensure algorithm stability on each task. Besides, a batch size of 256, $\sigma$ of $10^{-5}$, MVS of 1000, MF of 500, VUI of 1000 and $t$ of 10 for experience replay are employed. The testing of all tasks, sequentially executed on an NVIDIA A100 GPU, can be completed within a span of 200 hours.

### 4.1.2 RESULTS ON GOAL-DIRECTED TASKS

The results of RLDV on the 20 GuacaMol goal-directed tasks are shown in Table 1, compared with other official baselines and Reinvent, the most competitive baseline for *de novo* drug design as claimed by Gao et al. (2022).

Table 1: The evaluation results of RLDV and other baselines on the 20 GuacaMol goal-directed tasks.

| Tasks | dataset | Graph MCTS | SMILES GA | SMILES LSTM | Graph GA | Reinvent | RLDV |
|---|---|---|---|---|---|---|---|
| 1. Celecoxib rediscovery | 0.505 | 0.355 | 0.732 | **1.000** | **1.000** | **1.000** | **1.000** |
| 2. Troglitazone rediscovery | 0.419 | 0.311 | 0.515 | **1.000** | **1.000** | **1.000** | **1.000** |
| 3. Thiothixene rediscovery | 0.456 | 0.311 | 0.598 | **1.000** | **1.000** | **1.000** | **1.000** |
| 4. Aripiprazole similarity | 0.595 | 0.380 | 0.834 | **1.000** | **1.000** | **1.000** | **1.000** |
| 5. Albuterol similarity | 0.719 | 0.749 | 0.907 | **1.000** | **1.000** | **1.000** | **1.000** |
| 6. Mestranol similarity | 0.629 | 0.402 | 0.790 | **1.000** | **1.000** | **1.000** | **1.000** |
| 7. $C_{11}H_{24}$ | 0.684 | 0.410 | 0.829 | 0.993 | 0.971 | 0.999 | **1.000** |
| 8. $C_9H_{10}N_2O_2PF_2Cl$ | 0.747 | 0.631 | 0.889 | 0.879 | **0.982** | 0.877 | 0.956 |
| 9. Median molecules 1 | 0.334 | 0.225 | 0.334 | 0.438 | 0.406 | 0.434 | **0.448** |
| 10. Median molecules 2 | 0.351 | 0.170 | 0.380 | 0.422 | **0.432** | 0.395 | 0.425 |
| 11. Osimertinib MPO | 0.839 | 0.784 | 0.886 | 0.907 | 0.953 | 0.889 | **0.970** |
| 12. Fexofenadine MPO | 0.817 | 0.695 | 0.931 | 0.959 | 0.998 | **1.000** | **1.000** |
| 13. Ranolazine MPO | 0.792 | 0.616 | 0.881 | 0.855 | 0.920 | 0.895 | **0.939** |
| 14. Perindopril MPO | 0.575 | 0.385 | 0.661 | 0.808 | 0.792 | 0.764 | **0.810** |
| 15. Amlodipine MPO | 0.696 | 0.533 | 0.722 | 0.894 | 0.894 | 0.888 | **0.906** |
| 16. Sitagliptin MPO | 0.509 | 0.458 | 0.689 | 0.545 | **0.891** | 0.539 | 0.843 |
| 17. Zaleplon MPO | 0.547 | 0.488 | 0.413 | 0.669 | 0.754 | 0.590 | **0.770** |
| 18. Valsartan SMARTS | 0.259 | 0.040 | 0.552 | 0.978 | 0.990 | 0.895 | **0.993** |
| 19. deco hop | 0.933 | 0.590 | 0.970 | 0.996 | **1.000** | 0.994 | **1.000** |
| 20. scaffold hop | 0.738 | 0.478 | 0.885 | 0.998 | **1.000** | 0.990 | **1.000** |
| Total | 12.144 | 9.009 | 14.396 | 17.340 | 17.983 | 17.150 | **18.060** |

The results demonstrate that our RLDV algorithm outperforms all baselines in 17 out of the 20 tasks, with a notably higher total score than all baselines. This strongly attests to the superior performance of RLDV in *de novo* drug design tasks.

### 4.1.3 VISUALIZATION OF SUBSTRINGS VS. SUBSTRUCTURES

In order to validate whether the inclusion of higher-level tokens in the dynamic vocabulary is indeed beneficial for the generation of desirable molecules, we select the three rediscovery tasks from the GuacaMol benchmark. Each task involves the objective of generating a specific drug molecule. In all three tasks, our algorithm successfully rediscover the specified molecules (with a score of 1.0). We utilize the vocabulary at the conclusion of the algorithm's execution to tokenize the canonical SMILES strings of these three molecules. Subsequently, we visualize these substrings in 2D molecular graphs, as depicted in Figure 3.

The tokenization results of the three drug molecules are:

- Celecoxib: C, c1ccc(, -c2cc(C(F)(F)F)nn2-c2ccc(, S(N)(=O)=O)cc2)cc1

- Troglitazone: Cc1, c(C)c2c(c(C)c, 1O, )CC, C(C)(COc1ccc(CC3SC(=O)NC3=O)cc1)O2

- Thiothixene: CN1CC, N(, CC, /, C, =C2, /, c3ccccc3Sc3ccc(S(=O)(=O)N(C)C)cc32)CC1

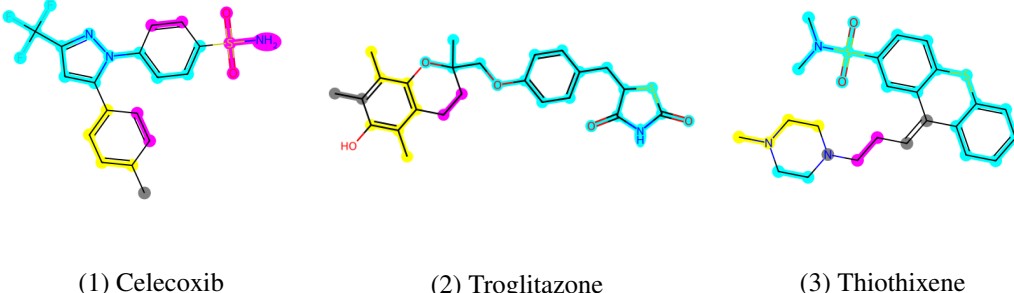

(1) Celecoxib      (2) Troglitazone      (3) Thiothixene

Figure 3: Visualization of the substructures corresponding to the tokens (SMILES substrings) on three drug molecules from the GuacaMol benchmark.

The results indicate that the dynamic vocabulary encompasses multiple substrings from the target molecule's SMILES representation, and these substrings exhibit a strong correspondence with molecular substructures. Most of the atomic-level tokens present in the target SMILES strings are covered by higher-level tokens in the dynamic vocabulary. Consequently, the utilization of the dynamic vocabulary significantly reduces the number of tokens required to generate the target SMILES strings. This observation suggests that the dynamic vocabulary effectively acquires structural knowledge of the target molecule and contributes to the efficiency of goal-directed molecular generation.

## 4.2 DESIGNING INHIBITORS AGAINST A SARS-CoV2 PROTEIN TARGET

The COVID-19 pandemic has inflicted significant global losses over the past few years. Its characteristics, such as high transmissibility and rapid mutation, underscore the critical importance of expedited drug development. The SARS-CoV-2 (Severe Acute Respiratory Syndrome Coronavirus 2) virus is the culprit behind this pandemic, and numerous protein target structures associated with this virus have been reported. Among these, the papain-like protease (PLPro) (Osipiuk et al., 2021), whose Protein Data Bank entry is 7JIR[1], is particularly intriguing. To validate the effectiveness of our RLDV algorithm in real-world *de novo* drug design, we apply it to the design of inhibitors against the 7JIR protein target.

We use the Quick Vina software (Hassan et al., 2017) for docking simulations, as the scoring function within RLDV. The binding affinity between small molecule inhibitors and protein targets typically falls within the range of -1.0 to -14.0 kcal/mol, with strong binding affinity typically being less than -12.0 kcal/mol. The top 100 high-scoring molecules designed by RLDV all meet this criterion as drug candidates.

Three exemplary instances with favorable properties are as follows:

- O=C(C(c1ccccc1)c1cccc(-c2cccc(-c3cccc(-c4cccc(-c5cccc(-c6ccccc6)c5)c4)c3)c2)c1)N1CCCC1
- O=C(c1ccnc(-c2cccc(-c3cccc(-c4cccc(-c5cccc(-c6ccccc6)c5)c4)c3)c2)c1)c1ccc2ccccc2c1
- CNc1nc(-c2cccc(-c3cccc(-c4cccc(-c5cccc(-c6cccc(C(=O)Nc7ccccc7)c6)c5)c4)c3)c2)cc(=O)[nH]1

In each SMILES string, a complete token present in the dynamic vocabulary is highlighted in purple, which corresponds to the substructures annotated in Figure 4. For instance, the first two molecules' strings correspond to five connected benzene rings, and the third molecule also contains five connected benzene rings. However, in the case of the third molecule, due to the presence of other atoms on both ends of the chain of benzene rings, it corresponds to a slightly shorter token at the SMILES level.

Figure 5 visualizes the binding modes of the three candidate inhibitors against the protein target. Here, we can observe that the substructure of five connected benzene rings aligns well with the

---

[1]https://www.rcsb.org/structure/7jir

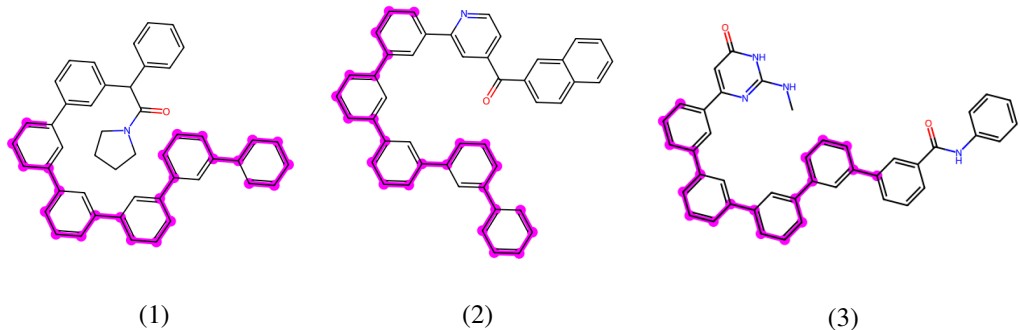

(1)                                (2)                                (3)

Figure 4: Visualization of common substrings in candidate inhibitors against the 7JIR target designed by RLDV.

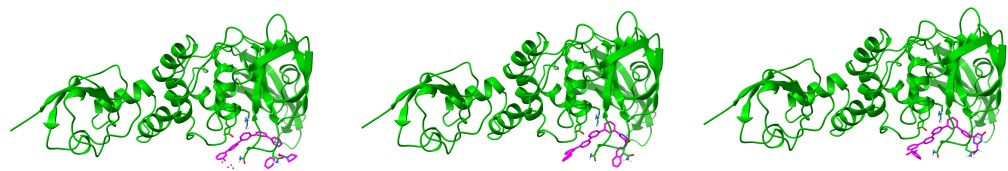

binding affinity: -13.4 kcal/mol   binding affinity: -12.0 kcal/mol   binding affinity: -12.2 kcal/mol
SA: 2.74                            SA: 2.33                            SA: 2.67

Figure 5: Visualization of the binding modes of designed candidate inhibitors against the 7JIR protein target, using the ChimeraX software (Pettersen et al., 2021).

7JIR target. This indicates that our dynamic vocabulary has indeed learned the crucial features for designing such drugs and explicitly preserving them in the form of SMILES sub-strings, which provide practical assistance in the design of candidate inhibitors.

It is also worth noting that the candidate molecules designed by RLDV exhibit favorable SA (synthetic accessibility) (Ertl & Schuffenhauer, 2009), which is advantageous for downstream drug development and production. This underscores the benefits commonly associated with fragment-based drug design methods and indirectly demonstrates the utility of the substructure information learned by RLDV.

## 5    CONCLUSION AND DISCUSSION

In this paper, we propose RLDV, a *de novo* drug design algorithm that leverages reinforcement learning and dynamic chemical vocabulary. By updating the vocabulary with high-frequency common substrings from SMILES strings of high-scoring molecules as tokens, RLDV exhibits superior sampling efficiency and task-specific interpretability. Our experiments on the GuacaMol benchmark and the design of inhibitors against SARS-CoV-2 protein target robustly demonstrate the advantages of our algorithm.

In the future, potential directions for improving our algorithm include:

1. Incorporating task-specific prior knowledge or expert experience to enhance the utilization of the dynamic vocabulary.

2. Facilitating diverse exploration by partitioning the dynamic vocabulary into distinct subsets.

3. Integrating 1D SMILES substrings with 2D graph substructures to design multi-modal dynamic models for molecular generation.

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
