# OpenReview forum: "De novo Drug Design using Reinforcement Learning with Dynamic Vocabulary"
_ICLR.cc/2024/Conference — ICLR 2024 Conference Withdrawn Submission_

### Official Review · Reviewer_449q · 2023-10-30

**Soundness:** 2 fair
**Presentation:** 2 fair
**Contribution:** 2 fair
**Rating:** 3
**Confidence:** 4

**Summary:**

This paper presents RLDV, which utilizes reinforcement learning (RL) and dynamic chemical vocabulary for de novo drug design. The authors update the atomic-level vocabulary using high-frequency common substrings from the SMILES strings of high-scoring molecules as tokens for the dynamic chemistry vocabulary. In the experiments, the authors validated RLDV using the GuacaMol benchmark, and the experimental results showed that the performance of RLDV improved to some extent.

**Strengths:**

Strengths:
- The authors proposed a dynamic chemical vocabulary-based algorithm for de novo goal-based drug design.
- The experimental results showed that the performance of RLDV improved to some extent.

**Weaknesses:**

Weaknesses:
- The authors proposed a chemical vocabulary that is dynamically updated to improve the effectiveness in RL. This dynamic updating is based on the commonly used sub-strings that occur with high frequency in the SMILES strings. Intuitively, the updated chemical vocabulary seems to be similar to a dictionary between the atomic level and the SELFIES level. Would the results also be improved if, when initially defining the chemical vocabulary (in the preprocessing phase), we counted high-frequency substrings based on SMILES and added them to the vocabulary?
- Compared to the SELFIES-based approaches, is the proposed SMILES-based dynamic chemical vocabulary approach improved? What are the advantages compared to other SELFIES-based and graph-based methods? Is there any improvement on the performance of the generated molecules (e.g., validity, novelty, and desired property scores)? The reviewer suggests the authors to conduct comparative experiments to verify the above questions.

**Questions:**

Questions:
- Please see the weaknesses.
- Why the GPT-2 model was chosen as a generator for SMILES strings? Why not select the latest GPT or BART models? The motivation should be clearly stated in the paper.
- The baseline models that the authors compared in their experiments were the most basic models. For a fair comparison, the reviewers suggest that the authors compare with more recent baseline models (2022 and 2023 models).
- Some typos:
    - Abbreviations should be defined at their first occurrence. For example, NLP and SMILES should be defined in paragraph 2.
    - RL, RLDV, MVS, and MF are defined repeatedly in the paper.
    - Inconsistent citation in section 2.2.

**Details Of Ethics Concerns:**

None.

---

### Official Review · Reviewer_uuoB · 2023-10-30

**Soundness:** 2 fair
**Presentation:** 2 fair
**Contribution:** 1 poor
**Rating:** 3
**Confidence:** 3

**Summary:**

This paper propose a dynamic vocabulary update algorithm for generating new molecules with chemical GPT. It fine-tunes a pre-trained GPT with REINFORCE to maximize a task-specific score while not deviating from the initial pretrained model. For each vocabulary update interval, the proposed method tokenizes top scoring molecules from memory and update the existing vocabulary with the new tokens. Empirically, it shows better performance than other baselines in GuacaMol dataset.

**Strengths:**

- The proposed method achieves better performance than other baselines in GuacaMol dataset.

**Weaknesses:**

- Since there is no ablation at all in this paper, it is hard to decide which component contributes to the performance improvement. First of all, it is difficult to say that dynamic vocabulary  is indeed useful for downstream tasks. What happens if we fine-tune the pre-trained GPT without any vocabulary update? Secondly, do we really need regularization that enforcing small difference between the prior and the agent in equation (1)?

- I am not convinced about how the vocabulary is updated. Although the generated molecule gets high score for a downstream task, some of the tokens from the molecules might be important for the downstream task. However, the proposed algorithm does not seem to handle this.

- The proposed method requires substantial amount of computational cost. The model is trained with A100 GPU for 200 hours.

- A missing baseline. MiCaM [1] which utilizes dynamic vocabulary update for molecule generation would be another relevant baseline.



---
[1] Geng, Zijie, et al. "De Novo Molecular Generation via Connection-aware Motif Mining." The Eleventh International Conference on Learning Representations. 2023.

**Questions:**

- Do we need labeled data for pre-training the GPT? The paper says that the model is trained with supervised learning. Do we need human-annotated labeled dataset for pre-training the GPT?

- What happens if we construct a new vocabulary by running the SMILES Pair Encoding on the target dataset and fine-tune the pre-trained GPT with union of existing vocabulary and the new vocabulary?

- Can we apply LoRA for fine-tuning the GPT?

---

### Official Review · Reviewer_MPZH · 2023-10-31

**Soundness:** 2 fair
**Presentation:** 2 fair
**Contribution:** 2 fair
**Rating:** 3
**Confidence:** 3

**Summary:**

This research presents RLDV, a method for creating new drugs through generative models. In contrast to existing approaches, RLDV utilizes a flexible chemical vocabulary in its reinforcement learning process. It analyzes successful molecular structures, identifies shared elements, and integrates them as new components. The experiments are performed on the widely used GuacaMol benchmark. Furthermore, the study's use in developing drugs that target proteins in SARS-CoV-2 highlights its significant practical importance.

**Strengths:**

- The paper is well-written and presents clearly.
- The paper demonstrates comprehensive related work.
- The experiments shows its practical applicability as shown in SARS-COV2 protein targeting experiments.

**Weaknesses:**

* Chemical Relevance of New Tokens:
    * The introduction of new tokens like cc, c(, C), and c2, as illustrated in Figure 2, does not seem to significantly enhance their chemical relevance compared to traditional atomic-level tokens such as C and c. A more detailed explanation regarding the choice and meaning of these new tokens is necessary to justify their incorporation into the vocabulary.
* Static vs. Dynamic Vocabulary:
    * The paper does not provide sufficient reasoning for opting for a dynamic vocabulary over a static one with a larger token set. Given that the chemical vocabulary is relatively constrained compared to the broader language domain, the benefits of employing a dynamic vocabulary in this context are not clearly articulated. It might be beneficial to elaborate on the advantages of dynamic token updating in relation to the specific chemical domain under consideration.
* Marginal Performance Gain:
    * The performance comparison presented in Table 1 reveals a marginal improvement with the proposed method (RLDV) achieving an aggregate performance of 18.060, slightly outperforming the second-best method (Graph GA) with a score of 17.983. The difference appears to be relatively minor.
* Novelty of Dynamic Vocabulary Usage:
    * The primary contribution of this paper revolves around the utilization of a dynamic vocabulary. However, the novelty of this approach is somewhat limited from my perspective. Further exploration of the unique advantages or applications enabled by this dynamic vocabulary within the realm of chemical analysis would strengthen the paper's originality and impact.

**Questions:**

See Weaknesses section

---

### Official Review · Reviewer_bE6S · 2023-11-01

**Soundness:** 1 poor
**Presentation:** 2 fair
**Contribution:** 1 poor
**Rating:** 3
**Confidence:** 4

**Summary:**

Through this paper, the authors aim to establish a SMILES-based generative model that overcomes the limitation of a fixed vocabulary. To accomplish this, the authors propose RLDV by constructing an RL model that uses a dynamic chemical vocabulary during RL iterations.

**Strengths:**

- The writing is easy to follow.

**Weaknesses:**

I will combine the *Weaknesses* section and the *Questions* section. My concerns and questions are as follows:

- In Section 3.2.1, the authors claim that SMILES pair encoding is one of the contributions. What is the difference between the proposed SMILES pair encoding and SmilesPE or simply applying BPE to SMILES strings?
- One of the main weaknesses of the paper is that the contribution (both conceptual and technical) is marginal. The proposed method seems like an additional trick rather than a machine learning-based strategy.
- One of the main weaknesses of the paper is that the experimental results are weak and insufficient. There is no appendix and the only experiment where the authors compare RLDV to existing methods is the GuacaMol benchmark.
   - Docking oracle (and real-world oracles for virtual screening) is expensive and the sample efficiency is very important in real-world drug discovery problems [1]. I highly recommend using the recently proposed PMO benchmark [1] instead of the GuacaMol benchmark, especially because the GuacaMol experiment is the only experiment that compares RLDV to existing methods in this paper.
   - The baselines in Table 1 are old and insufficient. More recent baselines like PS-VAE [2], MiCaM [3], FREED [4] should be included.
  - The GPT prior is pre-trained with the ChEMBL dataset. To be fair, an RLDV trained using only the GuacaMol dataset like the baselines in Table 1.
  - The proposed model is a generative model, but there is not enough visualization showing the molecules generated by RLDV in each task.
- The authors did not provide the codebase to reproduce the results.

---

**References:**

[1] Gao et al., Sample efficiency matters: a benchmark for practical molecular optimization, NeurIPS 2022.

[2] Kong et al., Molecule generation by principal subgraph mining and assembling, NeurIPS 2022.

[3] Geng et al., De novo molecular generation via connection-aware motif mining, ICLR 2023.

[4] Yang et al., Hit and lead discovery with explorative rl and fragment-based molecule generation, NeurIPS 2021.

**Questions:**

Please see the *Weaknesses* part for my main questions.

---

**Miscellaneous:**
- Section 2.1, paragraph Fragment-based drug design, *RS-VAE* -> *PS-VAE*